# Species Interactions and Nitrogen Use during Early Intercropping of Intermediate Wheatgrass with a White Clover Service Crop

**Shoujiao Li [1], Erik Steen Jensen [1], Nan Liu [2], Yingjun Zhang [2,\*] and Linda-Maria Dimitrova Mårtensson [1,\*]**

[1] Department of Biosystems and Technology, Swedish University of Agricultural Sciences, P.O Box 103, 230 53 Alnarp, Sweden; shoujiao.li@slu.se (S.L.); erik.steen.jensen@slu.se (E.S.J.)

[2] College of Grassland Sciences and Technology, China Agricultural University, Beijing 100193, China; liunan@cau.edu.cn

\* Correspondence: zhangyj@cau.edu.cn (Y.Z.); linda.maria.martensson@slu.se (L.-M.D.M.); Tel.: +86-13611272283 (Y.Z.); +46-722385816 (L.-M.D.M.)

**Abstract:** Perennial grain crops intercropped with legumes are expected to use nitrogen (N) resources efficiently. A pot experiment using the $^{15}$N isotope dilution method demonstrated interspecific competition and use of N from the soil and $N_2$ fixation in intermediate wheatgrass (*Thinopyrum intermedium* (Host) Barkworth & D.R. Dewey, IWG) and white clover (*Trifolium repens* L., WC) intercrops at five species-relative frequencies and four levels of inorganic N fertilizer in a replacement series design. The proportion of N in WC derived from the atmosphere increased from 39.7% in a sole crop to 70.9% when intercropped with IWG, and 10.1% N in IWG transferred from WC. Intermediate wheatgrass showed high fitness with maintained high total dry matter production at low relative frequencies. Decreasing IWG-relative frequency only increased dry matter and N accumulation of WC, resulting in increased amounts of $N_2$ fixed. Increased levels of N fertilization increased the proportion of N acquired from the fertilizer in IWG and WC but decreased the N fixed by WC and N absorbed by IWG from the soil. Our study indicates that WC supply sufficient fixed $N_2$ for IWG intercrop biomass yields under appropriate levels of soil N fertility and species-relative frequencies.

**Keywords:** symbiotic $N_2$ fixation; apparent transfer of N; intercropping advantages; interspecific interactions; intermediate wheatgrass; white clover; service crop

## 1. Introduction

Agriculture is under increasing pressure to improve productivity while limiting negative environmental impacts under the circumstances of climate change and population growth. The current global agriculture is dominated by the cultivation of annual crops, which may lead to many environmental problems, due to practices such as frequent tillage, reduced soil organic matter, and overuse of fertilizers and pesticides [1]. Perennial grain crops have been proposed by scientists to reduce these problems. They have extensive root systems and several years of permanent ground cover, which could increase water and nutrient use efficiency, soil organic matter, carbon sequestration, soil faunal diversity, and decrease tillage, soil erosion, and energy consumption [2]. Kernza is the first commercial perennial grain crop in the world, domesticated from the forage grass species intermediate wheatgrass (IWG) (*Thinopyrum intermedium* (Host) Barkworth & D.R. Dewey) [3]. The grain yield of IWG is currently much lower than that of annual wheat, but international breeding programs are working to increase yields. Despite the modest grain yields, organic and conventional farmers in France and the United States are inter-

ested in growing perennial grains for the reasons of increasing or maintaining farm profitability and improving soil health [4]. The capacity of IWG to reduce and prohibit nitrate leaching is confirmed by several studies [3,5]. Intermediate wheatgrass can be used to produce both grain and forage to provide additional economic benefits. Intercropping IWG with legumes has also been suggested to improve the feasibility of perennial grain cultivation [6]. The intercropping of cereals and legume service crops has been approved to produce greater yields, improve nutrient use efficiency, improve soil fertility through biological $N_2$ fixation, provide better lodging resistance, reduce pest incidence, improve forage quality, save synthetic fertilizer use, thus offering greater financial stability as compared to sole crops grown on the same amount of land [7,8].

The introduction of a legume service crop provides diverse ecosystem services to the intercropping systems [9,10]; however, it could also induce competition between the legume and cereal components. Choosing an appropriate companion legume service crop is important for establishing a stable mixture. Since perennial cereals are relatively new crops, limited research on intercropping with legumes has been performed. Hayes et al. [11] found that alternate rows of perennial wheat (*Thinopyrum* spp. × *Triticum aestivum* L.) lines and subterranean clover (*Trifolium subterraneum* L.) intercropping increased subterranean clover biomass and regeneration, and subterranean clover fixed sufficient N to maintain the N balance of a cropping system producing 1.5–2.0 t cereal grain ha$^{-1}$ each year. Intercropping of alfalfa (*Medicago sativa*) and IWG had similar IWG yields and nutrient acquisition and lower yield declines than the IWG sole crop fertilized with N in the Upper Midwestern USA [12]. Intercropping red clover (*Trifolium pratense* L.) with IWG consistently increased the nutritive value of the summer and fall forage of the first-year IWG and red clover stand in southern Wisconsin USA [13]. Our previous study showed that alfalfa was very aggressive when intercropped with IWG [14]. A similar result was also observed by Dick et al. [6], where alfalfa became dominant in alfalfa and IWG mixed pastures, thus, the IWG biomass was negatively influenced. However, IWG performed best when intercropping with white clover (*Trifolium repens* L., WC) compared to alfalfa or sweet clover (*Melilotus officinalis*) in the dual-purpose IWG system, because the prostrate growth habit and patterns of rooting depth of WC led to weak competition for light and nutrient [6,15]. The higher grain yields of IWG when grown in association with WC than when grown in monoculture were found in a field experiment at the Rodale Institute [15,16].

Furthermore, a higher rate of $N_2$ fixation [17] and N transfer [18–20] was reported from WC to neighboring plants compared to red clover or alfalfa. Literature data also show that up to 545 kg N ha$^{-1}$ year$^{-1}$ can be fixed by WC above-ground biomass in ungrazed northern temperate/boreal areas [17]. From 0 to 73% nitrogen could apparently be transferred from forage legumes to companion grasses in mixed stands, after one to four production years [21]. The $^{15}N$ isotope dilution method is one of the commonly used methods for the measurements of $N_2$ fixation. Soils often show slightly higher $^{15}N$ abundance than atmospheric $N_2$ does, due to the isotopic discrimination during biological, chemical, and physical processes [17]. This small difference can be utilized to distinguish between legume N derived from the soil and air, respectively. The higher $^{15}N$ abundance of soil derived N in legumes has been diluted by the low $^{15}N$ abundance of atmospheric $N_2$ as symbiotic $N_2$ fixation happened. A reference plant that relies only on soil nitrogen is used to estimate the $^{15}N$ abundance of soil N utilized by the legume. Artificially enriched $^{15}N$ fertilizer can be added to the soil to enlarge the difference between the $^{15}N$ composition of soil and the atmosphere [22]. Thus, the difference in $^{15}N$ abundances between the legume and the reference plant will be greater, allowing for precise estimations of $N_2$ fixation. A difficulty with this $^{15}N$ isotope dilution method is that the reference plant should have a similar pattern of N uptake as the legume and exploit the same soil N pool in order to obtain soil N of the same $^{15}N$ enrichment as the legume [17,22]. It is, therefore, important to make sure the legume and the reference plant utilize soil N from the same soil depth and at the same time, and the added $^{15}N$ is distributed evenly with soil depth and time.

In a legume and cereal intercropping system, the intercropping advantage can be influenced by both plant density and relative frequency of the intercrop components [23]. Relative frequency is the number of occurrences of a named species divided by the total occurrence of all species times one hundred [24]. Lithourgidis et al. [25] found that the relative yield total of the common vetch and oat mixtures exhibited an increasing trend as the common vetch proportion increased. Arlauskiene et al. [26] found that cereal aggressivity in the pea/barley, pea/oats, and pea/triticale intercrops depended on pea density, and in the pea/barley intercrops with an increasing number of pea plants, the competitive ratio of barley declined. Thus, the relative frequency of intercrop components could alter the outcome of competitive dynamics between component species and determine yields and production efficiency of cereal and legume intercropping systems. Up to now, how species-relative frequency might influence the interspecific competition, intercropping advantages, and yields of IWG and the service crop WC remains unknown.

Soil inorganic N concentration is also an important factor in determining intercropping advantages and interspecific interactions in a legume and cereal intercrop system [27]. Numerous studies have shown that the intercrop advantage is more evident on soils with low N availability, and it is significantly reduced by higher N input [26–29]. Intercropping advantages in cereal–legume intercrop are obtained, mainly due to the niche segregation for N resources between legumes and cereals [8,27] and potential N transfer from the legume to the cereal after some years [30].

Previous studies about N fertilization in the IWG cropping system mainly focused on the effects of N fertilizer on grain and biomass yields of IWG sole crops [31–33]. Jungers et al. [32] found that there was a quadratic response of IWG grain yield to increasing levels of N fertilizer where the optimal N rate range is 61 to 96 kg N ha$^{-1}$. Fernandez et al. [33] found that grain and biomass of IWG response to N fertilization were greatest in years 2 and 3. Tautges et al. [12] reported that N fertilization increased grain yield of IWG in year 2 but did not mitigate the decline in yields as stands aged. However, there is little published information on the effects of N fertilization on IWG and legume intercropping system. A legume service crop could fix atmospheric N$_2$ and supply N for the cereal intercrop, but a certain level of starter N is needed to overcome N stress until the nodules of legume are formed and capable for symbiotic N$_2$ fixation [34]. Increasing our understanding of how N fertilization impacts interspecific interaction and N use in early intercropping of IWG and WC is necessary to minimize the interspecific competition and maximize resource utilization in intercropping, thereby reducing the fertilizer inputs, minimizing environmental pollution, and optimizing agricultural productivity.

This study aimed to determine the effect of species-relative frequency and N fertilization on the competition for soil N sources, symbiotic N$_2$ fixation, dry matter yield, and intercropping advantage of IWG and WC intercropping systems during early growth. We hypothesized that (1) the symbiotic N$_2$ fixation will increase with the decrease in IWG-relative frequency, due to the decreased interspecific competition from IWG, (2) N fertilization will increase the interspecific competition at the advantages of IWG, and (3) higher N fertilizer levels will reduce intercropping advantages.

## 2. Materials and Methods

### 2.1. Greenhouse Experiment

The pot experiment was conducted in a greenhouse at China Agricultural University, Beijing, China, from 15 February to 4 July 2019. Supplemental light was supplied with high pressure sodium lamps (400 W, 100 μmol m$^{-2}$ s$^{-1}$) to give 16 h light and 8 h dark periods each day. The temperature was 26 °C during the day and 20°C during the night, and the air humidity was kept at 50% in the greenhouse. A loam soil was collected from the top 10 cm of a soil profile at the Shangzhuang Experimental Station (39° 59′ N, 116° 17′ E) of China Agricultural University. The chemical properties of the soil were: total N 537 mg kg$^{-1}$, nitrate N 11.0 mg kg$^{-1}$, ammonium N 2.08 mg kg$^{-1}$, total phosphorus 686 mg kg$^{-1}$,

available phosphorus 16.2 mg kg$^{-1}$, total potassium 11.6 g kg$^{-1}$, available potassium 75.5 mg kg$^{-1}$, pH$_{H20}$ 8.21, and soil organic matter 11.7 g kg$^{-1}$. Faba bean (*Vicia faba* L.) was the preceding crop in the field. Soil samples were sieved using a 2 mm sieve and homogenized. Pots with a diameter of 285 mm and a height of 265 mm (approx. 5 L) were filled with 10 kg soil and 4 L water added to each (70% water holding capacity).

The pot experiment followed a two-factor complete randomized design. The first factor was 4 levels of inorganic nitrogen fertilizer; N0, N1, N2, N3, corresponding to 0, 0.48, 0.96, and 1.44 g N pot$^{-1}$, which equaled approximately 0, 75, 150, and 225 kg N ha$^{-1}$, respectively. The second factor was 5 levels of species-relative frequency. In total, 16 plants per pot were planted according to a replacement series design, where intermediate wheatgrass (*Thinopyrum intermedium* (Host) Barkworth & D.R. Dewey, Cycle 3 from The Land Institute, a non-profit organization, Salina, Kansas, USA) (IWG) and white clover (*Trifolium repens* L.) (WC) were grown at five mixtures as 100% IWG (all 16 plants IWG), 75% IWG (12 plants IWG, 4 plants WC), 50% IWG (8 plants IWG, 8 plants WC), 25% IWG (4 plants IWG, 12 plants WC), and 0% IWG (0 plants IWG, 16 plants WC). Each treatment combination was replicated three times. There were 60 pots of plants (4 × 5 × 3) in this experimental design. The $^{15}$N-labeled ammonium nitrate ($^{15}$NH$_4$$^{15}$NO$_3$, 10.1% $^{15}$N) was used as the nitrogen fertilizer applied to the $^{15}$N-labeled treatments together with KCl. Two extra pots were supplied unlabeled N-fertilizer (ordinary KNO$_3$ and NH$_4$Cl) as a control to measure the background δ$^{15}$N value for the calculation of symbiotic N$_2$ fixation and apparent transfer of N. KCl was added to $^{15}$N-labeled treatments for keeping the form of the ions in fertilizers applied to treatments were as same as that of controls. Nitrogen fertilizer application was split into three applications to ensure the success of $^{15}$N isotope labeling, stabilize soil $^{15}$N enrichment by regular additions, and improve synchrony of N supply and demand. $^{15}$N-labeled fertilizer (N1, N2, N3) mother liquors were prepared using 6.84, 13.7, 20.5 g of $^{15}$NH$_4$$^{15}$NO$_3$ mixed with 6.37, 12.7, 19.1 g KCl and dissolved in 1 L distilled water, respectively. An aliquot of 66 mL mother liquor was diluted to 1 L and irrigated to each pot correspondingly to give 0, 0.16, 0.32, 0.48 g N pot$^{-1}$ at each application. For unlabeled controls, ordinary N fertilizer (N1, N2, N3) mother liquors were prepared using 5.75, 11.5, 17.3 g KNO$_3$ mixed with 3.05, 6.09, 9.14 g NH$_4$Cl and dissolved according to above. In total, 0, 0.48, 0.96, 1.44 g N pot$^{-1}$ was applied to N1, N2, N3 treatments after three applications of N fertilizer irrespective of $^{15}$N labeled treatments or unlabeled controls. Seeds of IWG were provided by the Swedish University of Agricultural Sciences. The seeds of WC were pre-inoculated with rhizobia bacteria (*Rhizobium leguminosarum biovar trifolii*). The agronomic practices and treatments are described in Table 1.

**Table 1.** The description of agronomic practices and treatments.

| Date | Agronomic Practices | Description |
|---|---|---|
| 16 February | Sowing | Seeds of intermediate wheatgrass (IWG) and white clover (WC) were sown simultaneously. |
| From 25 February | Watering | 500 mL water was irrigated to each pot weekly to keep soil moisture at 70% water holding capacity. |
| 3 to 10 March | Thinning | Five species-relative frequencies were formed by thinning seedlings. |
| From 19 March | Watering | 1 L water was irrigated once every four days to keep soil moisture at 70% water holding capacity. |
| 2 April | First N fertilizer application | $^{15}$NH$_4$$^{15}$NO$_3$ (10.1% $^{15}$N) and KCl were applied for $^{15}$N-labeled treatments, and KNO$_3$ and NH4Cl were applied for controls. |
| 6 April | Spraying pesticides | Pesticide thiosemicarbazide was sprayed on plants to control pest aphid. |
| 28 April | Second N fertilizer application | $^{15}$NH$_4$$^{15}$NO$_3$ (10.1% $^{15}$N) and KCl were applied for $^{15}$N-labeled treatments, and KNO$_3$ and NH4Cl were applied for controls. |
| 4 May | Spraying pesticides | Pesticide avermectin was sprayed on plants to control pest red spiders. |

| From 8 May | Watering | 2 L water was irrigated once every two days to keep soil moisture at 70% water holding capacity. |
| 6 June | Third N fertilizer application | $^{15}NH_4^{15}NO_3$ (10.1% $^{15}N$) and KCl were applied for $^{15}N$-labeled treatments, and $KNO_3$ and $NH_4Cl$ were applied for controls. |
| 16 June | Spraying pesticides | Pesticide bifenthrin was sprayed on plants to control pest pieris brassicae. |
| 2 July | Harvest and sampling | Shoots and roots of IWG and WC and soil samples were collected. |

### 2.2. *Plant and Soil Analyses*

#### 2.2.1. Dry Matter Yield

The harvest was done at the full-bloom stage of WC and the heading stage of IWG. Shoots were cut at the soil level and separated into IWG and WC shoots. The soil was removed from the pots, and roots were sifted out of the soil by using a sieve (2 mm). The roots of IWG and WC were separated according to their different shapes, colors, and the presence of nodules, after washing in tap water. All shoots and roots samples were oven-dried at 60 °C for 72 h for the measurements of the shoot and root dry matter.

#### 2.2.2. $^{15}N$ Abundance

Plant materials were ground to a fine powder by using two milling machines for the analyses of total nitrogen concentration and $^{15}N$ abundance. Plant samples were sent to the Institute of Agricultural Resources and Regional Planning, Chinese Academy of Agricultural Sciences, for isotope ratio mass spectrometry analyses of nitrogen isotopes.

#### 2.2.3. Soil Inorganic N and pH

After storage at −20 °C, 50 mL 1 mol $L^{-1}$ KCl was added to 12 g of fresh soil in 100 mL plastic tubes and shaken for 30 min at 250 rpm. The soil inorganic N concentration of extracts was analyzed using a continuous flow mass spectrometer (SEAL AutoAnalyzer 3) by the UV-absorbance spectrophotometer method [35]. Soil water content was measured based on the gravimetric method for the calculation of soil inorganic N concentrations. Soil pH was measured using a pH meter on the filter extract of 10 g air dried soil extracted in 50 mL distilled water after shaking for 30 min at 275 rpm.

### 2.3. *Nitrogen Acquisition*

#### 2.3.1. $N_2$ Fixation and N Transfer

The proportion of N derived from the atmosphere of WC shoot or root (%NA$_{SHOOT or ROOT}$, %) was calculated following the $^{15}N$ isotope dilution method [36,37] using Equation (1).

$$\%NA_{SHOOT\ or\ ROOT} = (1 − [atom\%\ ^{15}N\ excess_{WC}/atom\%\ ^{15}N\ excess_{IWG\ SOLE}]) \times 100 \qquad (1)$$

The term "atom% $^{15}N$ excess" reflects the $^{15}N$ enrichment above the background levels of unlabeled growth environments, i.e., the atom% $^{15}N$ excess is atom% $^{15}N$ of labeled samples (three replicates) minus the atom% $^{15}N$ of unlabeled controls (two replicates). Here, the atom% $^{15}N$ excess$_{WC}$ indicates the atom% $^{15}N$ excess of the legume crop WC, and the atom% $^{15}N$ excess$_{IWGSOLE}$ indicates the atom% $^{15}N$ excess of the non-leguminous IWG sole crop. The calculation of %NA was done for the shoots and roots of WC separately, as well as for intercrops and sole crops of WC under each N fertilizer rate. That is, the atom% $^{15}N$ excess of shoots and roots of WC in intercrops was used to calculate the %NA of shoots and roots of intercropped WC for each N fertilizer rate, and atom% $^{15}N$ excess of shoots and roots of WC sole crops was used to calculate the %NA of shoots and roots of sole cropped WC for each fertilizer level, while always the atom% $^{15}N$ excess of shoots and

roots of sole cropped IWG was used as the non-fixing reference to calculate %NA at a given N level.

The amount of N fixed by WC shoot or root ($NFIX_{SHOOT\ or\ ROOT}$, g $pot^{-1}$) was determined using Equation (2) [36], where $Y_{WC}$ is the dry matter yield of WC shoot or root, %$N_{WC}$ is the N concentration of WC shoot or root.

$$NFIX_{SHOOT\ or\ ROOT} = Y_{WC} \times \%N_{WC}/100 \times \%NA_{SHOOT\ or\ ROOT}/100 \qquad (2)$$

The proportion of fixed N of WC whole plant (%$NA_{WC\ TOTAL}$, %) was calculated using Equation (3), where $NFIX_{SHOOT}$ and $NFIX_{ROOT}$ indicate the amount of N fixed by WC shoot and root, respectively, while $N_{SHOOT}$ and $N_{ROOT}$ indicate the N accumulated in WC shoots and roots, respectively. Thereafter, the accumulation of N in WC was calculated by multiplying the N concentration of WC by the dry matter of WC.

$$\%NA_{WC\ TOTAL} = (NFIX_{SHOOT} + NFIX_{ROOT})/(N_{SHOOT} + N_{ROOT}) \times 100 \qquad (3)$$

The percentage of N in IWG intercrops apparently transferred from WC intercrops was calculated by comparing $^{15}N$ enrichment in IWG mixed intercrops versus IWG sole crop at a given N level, following the $^{15}N$ isotope dilution method [36,37]. The percentage of N apparently transferred to IWG shoot and root (%$NT_{SHOOT\ or\ ROOT}$, %) was calculated separately using Equation (4) [37], where atom% $^{15}N$ excess$_{IWGMIX}$ indicates the atom% $^{15}N$ excess of IWG mixed intercrops, and atom% $^{15}N$ excess$_{IWGSOLE}$ indicates the atom% $^{15}N$ excess of IWG sole at each N fertilizer level.

$$\%NT_{SHOOT\ or\ ROOT} = (1 - [atom\%\ ^{15}N\ excess_{IWG\ MIX}/atom\%\ ^{15}N\ excess_{IWG\ SOLE}]) \times 100 \qquad (4)$$

Then, the amount of N apparently transferred to IWG shoot or root ($NT_{SHOOT\ or\ ROOT}$, g $pot^{-1}$) was determined for each IWG intercrop under each relative frequency and N fertilizer rate using Equation (5) [37], where $Y_{IWG}$ is the dry matter yield of IWG shoot or root, and %$N_{IWG}$ is the N concentration of IWG shoot or root at a given IWG frequency and given N fertilizer rate.

$$NT_{SHOOT\ or\ ROOT} = Y_{IWG} \times \%N_{IWG}/100 \times \%NT_{SHOOT\ or\ ROOT}/100 \qquad (5)$$

The percentage of N apparently transferred to IWG whole plant (%$NT_{IWG\ TOTAL}$, %) was calculated using Equation (6), where $NT_{SHOOT}$ represents the amount of N transferred to IWG shoot, and $NT_{ROOT}$ represents the amount of N transferred to IWG root, $N_{SHOOT}$ and $N_{ROOT}$ represent the amount of N accumulation of IWG shoot and root, respectively.

$$\%NT_{IWG\ TOTAL} = (NT_{SHOOT} + NT_{ROOT})/(N_{SHOOT} + N_{ROOT}) \times 100 \qquad (6)$$

### 2.3.2. N Derived from Fertilizer and Soil

The proportion of N derived from the fertilizer (%NF) was estimated by comparing $^{15}N$ enrichment in the plant (IWG and WC) versus $^{15}N$ enrichment in the labeled fertilizer at each IWG-relative frequency and N fertilizer rate using Equation (7) [34,38]. The atom% $^{15}N$ excess of IWG was used for calculating the %NF of IWG, and atom% $^{15}N$ excess of WC was used for calculating the %NF of WC. The same atom% $^{15}N$ excess of N fertilizer was used for the calculation of %NF in IWG or WC at a given N fertilizer rate.

$$\%NF = (atom\%\ ^{15}N\ excess_{IWG\ or\ WC}/atom\%\ ^{15}N\ excess_{FERTILIZER}) \times 100 \qquad (7)$$

The proportion of N derived from the unlabeled soil (%NS) was calculated with the assumption that N accumulated in WC and IWG arise from fertilizer and soil in both cases, while also from the atmosphere for WC and from transfer in IWG [39,40] (Equation (8) for WC and Equation (9) for IWG). The %NF represents the proportion of N derived from fertilizer, %NS the proportion of N derived from soil, %NA the proportion of N derived from the atmosphere, and %NT the proportion of N transferred from WC to IWG.

$$\%NF + \%NS + \%NA = 100\% \tag{8}$$

$$\%NF + \%NS + \%NT = 100\% \tag{9}$$

The amount of N derived from fertilizer in plant shoot and root were calculated by multiplying the %NF of the plant shoot or root by the dry matter of shoot or root, which were summed up to give the amount of N derived from fertilizer in the whole plant. The %NF of the whole plant was calculated by dividing the amount of N derived from the fertilizer of whole plants by the N accumulation of whole plants and multiplying by 100. The same method was used to calculate the %NS of the whole plant.

Fertilizer N recovery (%) by the crop was calculated for each treatment by the equation presented by IAEA [38] and Jørgensen et al. [36] (Equation (10)). The %NF, total $N_{IWG \text{ or } WC}$, and total $N_{FERTILIZER}$ are derived from the calculations above.

$$\text{Recovery} = (\%NF \times \text{total } N_{IWG \text{ or } WC}/\text{total } N_{FERTILIZER}) \times 100 \tag{10}$$

The same amount of N fertilizer was used for calculating both IWG and WC fertilizer N recoveries. Then, the total recovery for the whole cropping systems was calculated as the sum of N recoveries of IWG and WC.

*2.4. Intercropping Advantages and Interspecific Interactions*

2.4.1. Relative Yield Total

The relative advantage of mixed intercropping compared to sole cropping was estimated by the relative yield total (RYT) [27] (Equation (11)), where $Y_{IWG \text{ MIX}}$ and $Y_{WC \text{ MIX}}$ indicate the dry matter yields of IWG and WC mixed intercrops per pot, $Y_{IWG \text{ SOLE}}$ and $Y_{WC \text{ SOLE}}$ indicate the mean of dry matter yields of five pots with IWG and WC sole crops under the same N fertilizer level.

$$\text{RYT} = (Y_{IWG \text{ MIX}}/Y_{IWG \text{ SOLE}}) + (Y_{WC \text{ MIX}}/Y_{WC \text{ SOLE}}) \tag{11}$$

An RYT larger than one indicates an advantage for intercropping compared to sole cropping. An RYT less than one indicates an advantage for sole cropping, while an RYT of one indicates no advantages from mixed intercropping compared to sole cropping.

2.4.2. Competitive Ratio

The competitive ratio of IWG ($CR_{IWG}$) was used as an indicator to evaluate the competitive ability of IWG relative to WC, and the $CR_{WC}$ was used to evaluate the competitive ability of WC relative to IWG. The competitive ratio represents the ratio of individual RYTs of the two component crops and takes into account the proportion of the crops in which they are initially sown [41] (Equations (12) and (13)).

$$CR_{IWG} = (Y_{IWG \text{ MIX}}/Y_{IWG \text{ SOLE}} \times \text{IRF})/(Y_{WC \text{ MIX}}/Y_{WC \text{ SOLE}} \times [1\text{-IRF}]) \tag{12}$$

$$CR_{WC} = 1/CR_{IWG} \tag{13}$$

The $Y_{IWG \text{ MIX}}$ and $Y_{WC \text{ MIX}}$ represent the dry matter yields of IWG and WC mixed intercrops per pot, $Y_{IWG \text{ SOLE}}$ and $Y_{WC \text{ SOLE}}$ represent the dry matter yields of IWG and WC sole crops per pot. IRF is the IWG-relative frequency, which equals the initial sown proportion of IWG intercrops, and 1-IRF is the WC-relative frequency, which equals the initial sown proportion of WC intercrops. When $CR_{IWG}$ is greater than one, the competitive ability of IWG is higher than WC in mixed intercrops. Contrarily, when the $CR_{WC}$ is greater than one, the competitive ability of WC is higher than IWG.

*2.5. Statistical Analyses*

The main effects of N fertilizer and species-relative frequency, and the N fertilizer × species-relative frequency interaction were assessed using analysis of variance (two-way

ANOVA), performed by the general linear model (GLM) in IBM SPSS statistics 23.0. When the effect of the treatments was found to be significant (*F*-tests, *p*-value < 0.05), means were compared using Tukey's HSD test at $\alpha$ = 0.05. All the measured variables fulfilled the assumptions of normal distribution and homogenous variances. Three replicates were used in the analysis of variance and calculation of means and standard error for all responsible variables.

## 3. Results

### 3.1. IWG and WC Dry Matter Yield

The shoot and root dry matter productions of most IWG were not significantly influenced by a decrease in IWG-relative frequency within each N fertilizer level, except for the 25% IWG, which has lower yields than IWG sole crop at N2 and N3 (Figure 1). Within a specific IWG-relative frequency, the IWG shoot dry matter increased with increased N supply, reaching apparently a maximum at the N3 nitrogen fertilizer level. The IWG root dry matter was higher at the N1 fertilization level compared to the N0, but additional N did not increase root dry matter further. The shoot and root dry matters of WC increased as the level of IWG-relative frequency decreased, reaching a maximum in the sole crop WC, within each N fertilizer level (Figure 1). Nitrogen fertilization did not significantly influence the WC shoot and root dry matter productions. The shoot and root dry matters of WC were always lower than the dry matter of IWG regardless of species-relative frequencies and N fertilizer levels in intercrops.

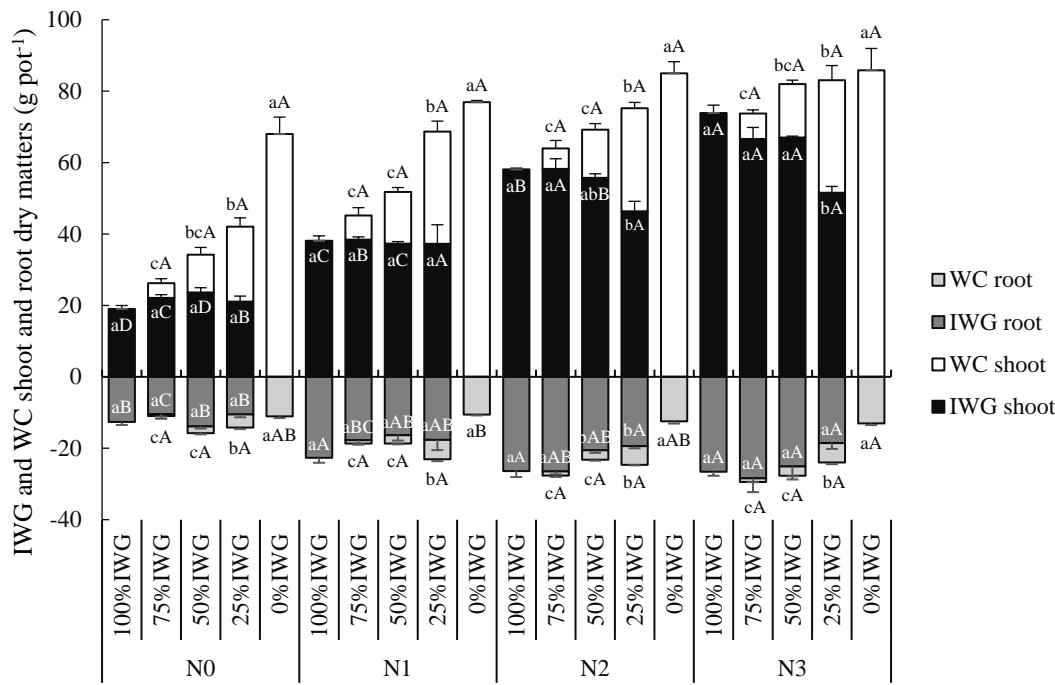

**Figure 1.** Shoot and root dry matter of intermediate wheatgrass (IWG) and white clover (WC) per pot under four N fertilizer levels (N0, N1, N2, and N3) and five IWG-relative frequencies (IRF) (100% IWG, 75% IWG, 50% IWG, 25% IWG, 0% IWG). The absolute values of numbers on the negative side of the *Y*-axis are the root dry matter of IWG and WC. Different lower-case letters indicate significant differences at *p* < 0.05 among IRF under the same N level, and different upper-case letters indicate significant differences at *p* < 0.05 among N levels under the same IRF (Tukey's post hoc test).

The total dry matters of IWG and WC intercrops at 50 and 25% IWG were higher than that of the IWG sole crop but lower than WC sole crop at N0 (Table 2). The total dry matter of IWG and WC intercrop at 25% IWG was higher than IWG sole crop and similar to WC sole crop at N1 and N2. There was no significant difference between intercrops and sole crops at N3. Within a specific IWG-relative frequency, the total dry matter of IWG and

WC intercrops increased with the increase in the N fertilizer level. The IWG root/shoot ratio was significantly reduced with increasing N fertilizer level at 100% IWG and 50% IWG, while the WC root/shoot ratio at 50% IWG was significantly higher than the WC sole crop at N0 fertilizer level.

**Table 2.** The total dry matter yields per pot, root/shoot ratio of intermediate wheatgrass (IWG) and white clover (WC), relative yield total (RYT), and the competitive ratio of IWG (CR$_{IWG}$) and WC (CR$_{WC}$) under five relative frequencies of IWG (IRF) and four N fertilizer levels (N). Data are presented as mean ± standard error (*n* = 5). F-statistics and significance from ANOVA are reported below the treatment means.

| N | IRF | Total Yields (g Pot$^{-1}$) | IWG Root/Shoot | WC Root/Shoot | RYT | CR$_{IWG}$ | CR$_{WC}$ |
|---|---|---|---|---|---|---|---|
| N0 | 100%IWG | 32.5 ± 1.13cD | 0.63 ± 0.00A | | | | |
| | 75%IWG | 37.6 ± 1.15cD | 0.51 ± 0.11A | 0.15 ± 0.01b | 1.05 ± 0.04aA | 7.04 ± 2.02A | 0.17 ± 0.05a |
| | 50%IWG | 51.0 ± 2.59bD | 0.57 ± 0.03A | 0.18 ± 0.01ab | 1.31 ± 0.03aA | 8.16 ± 1.13A | 0.13 ± 0.02a |
| | 25%IWG | 55.4 ± 1.99bB | 0.51 ± 0.07A | 0.19 ± 0.01a | 1.30 ± 0.01aA | 9.43 ± 1.48A | 0.11 ± 0.02a |
| | 0%IWG | 79.0 ± 5.23aA | | 0.15 ± 0.01b | | | |
| N1 | 100%IWG | 60.8 ± 2.34cC | 0.60 ± 0.03A | | | | |
| | 75%IWG | 63.8 ± 2.76cC | 0.46 ± 0.04A | 0.15 ± 0.02a | 1.07 ± 0.03aA | 5.12 ± 2.36A | 0.28 ± 0.09a |
| | 50%IWG | 70.5 ± 0.85bcC | 0.44 ± 0.03AB | 0.16 ± 0.01a | 1.13 ± 0.02aB | 4.84 ± 0.58A | 0.21 ± 0.03a |
| | 25%IWG | 91.8 ± 8.82aA | 0.47 ± 0.02A | 0.17 ± 0.00a | 1.39 ± 0.15aA | 6.77 ± 1.08A | 0.16 ± 0.03a |
| | 0%IWG | 87.5 ± 0.57abA | | 0.14 ± 0.00a | | | |
| N2 | 100%IWG | 84.5 ± 1.37bB | 0.45 ± 0.03B | | | | |
| | 75%IWG | 91.6 ± 1.70abB | 0.46 ± 0.01A | 0.21 ± 0.01a | 1.10 ± 0.02aA | 6.47 ± 2.57A | 0.22 ± 0.09a |
| | 50%IWG | 92.4 ± 2.05abB | 0.37 ± 0.01B | 0.20 ± 0.02a | 1.10 ± 0.02aB | 5.56 ± 0.74A | 0.19 ± 0.02a |
| | 25%IWG | 99.9 ± 3.70aA | 0.42 ± 0.03A | 0.18 ± 0.01a | 1.16 ± 0.04aA | 6.61 ± 0.41A | 0.15 ± 0.01a |
| | 0%IWG | 97.4 ± 2.85aA | | 0.15 ± 0.01a | | | |
| N3 | 100%IWG | 100 ± 1.34aA | 0.36 ± 0.03B | | | | |
| | 75%IWG | 103 ± 3.06aA | 0.43 ± 0.08A | 0.15 ± 0.03a | 1.03 ± 0.03aA | 4.00 ± 0.59A | 0.26 ± 0.04a |
| | 50%IWG | 110 ± 3.69aA | 0.38 ± 0.06B | 0.17 ± 0.01a | 1.10 ± 0.04aB | 5.23 ± 0.39A | 0.19 ± 0.01a |
| | 25%IWG | 107 ± 4.36aA | 0.36 ± 0.04A | 0.18 ± 0.02a | 1.07 ± 0.04aA | 5.83 ± 0.79A | 0.18 ± 0.02a |
| | 0%IWG | 98.9 ± 6.54aA | | 0.15 ± 0.01a | | | |
| **F-statistic** | | | | | | | |
| **Source of variation** | | | | | | | |
| N | | 213 *** | 10.5 *** | 2.72 | 5.39 ** | 3.03 * | 2.06 |
| IRF | | 26.4 *** | 2.16 | 5.04 ** | 10.2 ** | 1.33 | 3.60 * |
| N*IRF | | 6.77 *** | 1.06 | 1.04 | 2.99 * | 0.20 | 0.11 |

Notes: Different lower-case letters indicate significant differences at *p* < 0.05 among IRF under the same N level, and different upper-case letters indicate significant differences at *p* < 0.05 among N levels under the same IRF (Tukey's post hoc test). IRF means the species-relative frequency of IWG. Asterisks indicate significant differences, where * indicates *p* < 0.05, ** *p* < 0.01, and *** *p* < 0.001.

### 3.2. Intercropping Advantages and Interspecific Interactions

The relative yield total (RYT) did not differ among the IWG-relative frequencies under all N fertilizer conditions (Table 2). Nitrogen fertilization did not have any effect on RYT at 75% IWG and 25% IWG. However, at 50% IWG, RYT was higher under the N0 fertilization level than under the N1, N2, and N3 levels. The competitive ratio of IWG (CR$_{IWG}$) was larger than one under all treatments, while the competitive ratio of WC (CR$_{WC}$) was less than one.

### 3.3. The Proportion of N Derived from Different N Sources

Both the proportion of N derived from soil (%NS) and fertilizer (%NF) of IWG had a tendency to decrease with the decrease in IWG-relative frequency at N1 and N2 fertilizer

levels and remained unchanged at N3 (Figure 2). The proportion of apparent transfer N (%NT) was unaffected by the decrease in IWG-relative frequency at all N fertilizer levels. Within a specific IWG-relative frequency, the %NS decreased with the increase in N fertilizer rates, while the %NF increased, and %NT remained unchanged.

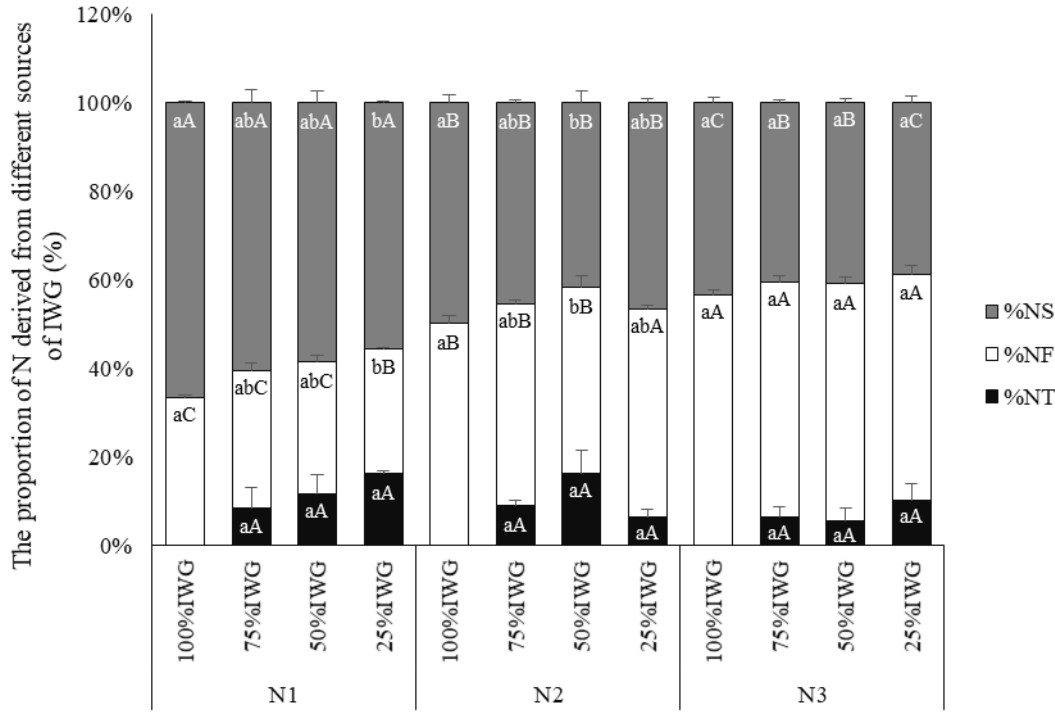

**Figure 2.** The proportion of N derived from the soil (%NS), fertilizer (%NF), and apparently transferred from white clover (%NT) in intermediate wheatgrass (IWG) whole plant under three N fertilizer levels (N1, N2, and N3) and four IWG-relative frequencies (IRF) (100% IWG, 75% IWG, 50% IWG, 25% IWG). Different lower-case letters indicate significant differences at $p < 0.05$ among IRF under the same N level, and different upper-case letters indicate significant differences at $p < 0.05$ among N levels under the same IRF (Tukey's post hoc test).

Both %NS and %NF of WC intercrops are lower than WC sole crops at all N fertilizer levels (Figure 3). Conversely, the proportions of N derived from $N_2$ fixation (%NA) in WC intercrops are higher than in WC sole crops at all N fertilizer levels. There were no significant differences among different WC intercrops (75% IWG, 50% IWG, and 25% IWG), either of %NA, %NF, or %NS. Within a specific IWG-relative frequency, the %NS of WC remained constant with the increase in N fertilizer levels, while the %NF increased, and %NA decreased sharply.

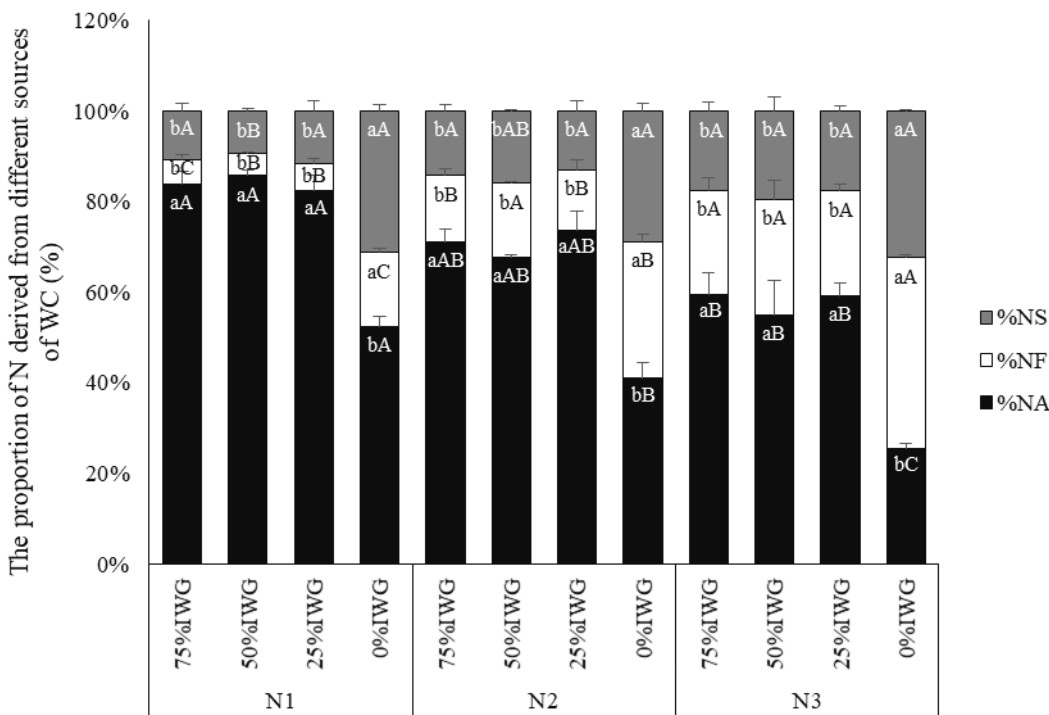

**Figure 3.** The proportion of N derived from the soil (%NS), fertilizer (%NF), and atmosphere (%NA) in white clover (WC) whole plant under three N fertilizer levels (N1, N2, and N3) and four IWG-relative frequencies (IRF) (75% IWG, 50% IWG, 25% IWG, and 0% IWG). Different lower-case letters indicate significant differences at *p* < 0.05 among IRF under the same N level, and different upper-case letters indicate significant differences at *p* < 0.05 among N levels under the same IRF (Tukey's post hoc test).

### 3.4. Nitrogen Accumulation and Transfer

#### 3.4.1. Nitrogen Accumulation

The total N accumulations of IWG and WC intercrops at 50 and 25% IWG were higher than that of the IWG sole crop but lower than WC sole crop at N0 (Table 3). The total N accumulations of intercrops at 50 and 25% IWG were higher than that of the IWG sole crop, and only the N accumulation at 25% IWG was similar to WC sole crop at N1 and N2. The total N accumulation of intercrop at 25% IWG was higher than that of the IWG sole crop and similar to WC sole crop at N3. The total N accumulation of intercrops tended to increase with the decrease in IWG-relative frequency from 75% IWG to 25% IWG. Within a specific species-relative frequency, the total N accumulations increased with the increase in the N fertilizer level except for the total N accumulation of WC sole crop. Nitrogen accumulation of IWG increased with the increase in N fertilizer level under the same IWG-relative frequency, reaching a maximum at the highest N fertilizer level N3. Nitrogen accumulation of WC increased with the decrease in IWG-relative frequency under all fertilizer levels, reaching a maximum at sole crops (0% IWG).

**Table 3.** The amount of total N accumulation per pot (Total N), N accumulation of intermediate wheatgrass (IWG N) and white clover whole plants (WC N), $N_2$ fixation of WC, and apparent transfer of N to IWG under five relative frequencies of IWG (IRF) and four N fertilizer levels (N). Data are presented as mean ± standard error. F-statistics and significance from ANOVA are reported below the treatment means.

| N | IRF | Total N (g Pot⁻¹) | IWG N (g Pot⁻¹) | WC N (g Pot⁻¹) | N₂ Fixation (g Pot⁻¹) | N Transfer (g Pot⁻¹) |
|---|---|---|---|---|---|---|
| N0 | 100%IWG | 0.37 ± 0.03dD | 0.37 ± 0.03D | | | |
| | 75%IWG | 0.47 ± 0.06cdC | 0.36 ± 0.02C | 0.09 ± 0.03cA | ND | ND |
| | 50%IWG | 0.72 ± 0.05bcD | 0.46 ± 0.01D | 0.24 ± 0.03bcA | ND | ND |

| | | | | | | |
|---|---|---|---|---|---|---|
| | 25%IWG | 0.91 ± 0.04bC | 0.42 ± 0.01C | 0.49 ± 0.05bA | ND | ND |
| | 0%IWG | 1.56 ± 0.10aA | | 1.56 ± 0.10aA | ND | ND |
| N1 | 100%IWG | 0.70 ± 0.02cC | 0.70 ± 0.02C | | | |
| | 75%IWG | 0.80 ± 0.05bcB | 0.65 ± 0.02B | 0.15 ± 0.05dA | 0.12 ± 0.04cA | 0.06 ± 0.03 |
| | 50%IWG | 1.03 ± 0.02bC | 0.70 ± 0.01C | 0.33 ± 0.03cA | 0.28 ± 0.02cA | 0.08 ± 0.03 |
| | 25%IWG | 1.43 ± 0.12aB | 0.74 ± 0.10B | 0.69 ± 0.05bA | 0.57 ± 0.05bA | 0.12 ± 0.01 |
| | 0%IWG | 1.71 ± 0.01aA | | 1.71 ± 0.01aA | 0.89 ± 0.04aA | |
| N2 | 100%IWG | 1.08 ± 0.03cB | 1.08 ± 0.03B | | | |
| | 75%IWG | 1.33 ± 0.02bcA | 1.20 ± 0.06A | 0.13 ± 0.05cA | 0.09 ± 0.04cA | 0.11 ± 0.02 |
| | 50%IWG | 1.35 ± 0.04bB | 1.05 ± 0.05B | 0.30 ± 0.03cA | 0.20 ± 0.02cA | 0.18 ± 0.06 |
| | 25%IWG | 1.69 ± 0.08aAB | 1.04 ± 0.05A | 0.64 ± 0.04bA | 0.48 ± 0.05bA | 0.07 ± 0.02 |
| | 0%IWG | 1.81 ± 0.08aa | | 1.81 ± 0.08aa | 0.75 ± 0.09aA | |
| N3 | 100%IWG | 1.40 ± 0.03bA | 1.40 ± 0.03A | | | |
| | 75%IWG | 1.47 ± 0.04bA | 1.33 ± 0.05A | 0.14 ± 0.02cA | 0.08 ± 0.00bA | 0.08 ± 0.03 |
| | 50%IWG | 1.65 ± 0.05abA | 1.34 ± 0.04A | 0.31 ± 0.02cA | 0.17 ± 0.03bA | 0.07 ± 0.04 |
| | 25%IWG | 1.84 ± 0.07aA | 1.17 ± 0.03A | 0.67 ± 0.08bA | 0.40 ± 0.06aA | 0.12 ± 0.04 |
| | 0%IWG | 1.84 ± 0.08aA | | 1.84 ± 0.08aA | 0.47 ± 0.04aB | |
| **F-statistic** | | | | | | |
| **Source of variation** | | | | | | |
| N | | 192 *** | 369 *** | 5.91 ** | 16.6 *** | 0.78 |
| IRF | | 133 *** | 1.03 | 722 *** | 106 *** | 0.50 |
| N*IRF | | 6.40 *** | 3.12 ** | 1.00 | 3.65 * | 1.75 |

Notes: Different lower-case letters indicate significant differences at $p < 0.05$ among IRF under the same N level, and different upper-case letters indicate significant differences at $p < 0.05$ among N levels under the same IRF (Tukey's post hoc test). IRF means the species-relative frequency of IWG. ND means not determined. Asterisks indicate significant differences, where * indicate $p < 0.05$, ** $p < 0.01$, and *** $p < 0.001$.

### 3.4.2. N₂ Fixation and Apparent Transfer of N

The amount of $N_2$ fixed by WC sole crop was higher than WC intercrops at all N fertilizer levels, and the $N_2$ fixation of WC intercrops tended to increase with the decrease in IWG-relative frequency, with no differences found among 75% IWG and 50% IWG (Table 3). Within a specific relative frequency, only $N_2$ fixation of WC sole crop was lower at N3 than N1 and N2 fertilizer levels. The amount of apparent N transfer from WC to IWG was unaffected by species-relative frequency or N fertilizer.

### 3.5. Soil Inorganic N Concentration after Harvest

The total fertilizer N recovery is affected by N fertilizer levels (Table 4). The total fertilizer N recovery in the 75% IWG treatment was higher at the N2 fertilizer level than N1. The fertilizer N recovery of IWG decreased with the decrease in IWG-relative frequency at N2 and N3 fertilizer levels. Within a specific IWG-relative frequency, the N recovery of 75% IWG was higher at N2 than the N1 fertilizer level. The fertilizer N recovery of WC was affected by species-relative frequency; the N recovery of WC increased with the decrease in IWG-relative frequency. The soil mineral N concentration tended to increase with the increase in N fertilizer level under treatments of 100% IWG, 50% IWG, and 0% IWG. No significant differences were detected among IWG-relative frequencies irrespective of the N fertilizer level. Soil pH of IWG and WC intercrops at 50 and 25% IWG were lower than IWG and WC sole crops at N0. Soil pH of WC sole crop was higher than mixed intercrops and IWG sole crop at N1 fertilizer level, with no difference between the mixed intercrops and the IWG sole crop. Within a specific IWG-relative frequency, soil pH decreased with the increase in N fertilizer levels at 100% IWG and 75% IWG.

**Table 4.** The total fertilizer N recovery (Recovery total), N recovery of intermediate wheatgrass (Recovery IWG) and white clover (Recovery WC), the concentration of soil mineral N, and pH value under five relative frequencies of IWG (IRF) and four N fertilizer levels (N). Data are presented as mean ± standard error. F-statistics and significance from ANOVA are reported below the treatment means.

| N | IRF | Recovery Total (%) | Recovery IWG (%) | Recovery WC (%) | Soil Mineral N (mg kg⁻¹) | pH |
|---|---|---|---|---|---|---|
| N0 | 100%IWG | | | | 3.14 ± 0.22aB | 8.16 ± 0.02aA |
| | 75%IWG | | | | 3.44 ± 0.24aA | 8.10 ± 0.04abA |
| | 50%IWG | | | | 2.86 ± 0.32aB | 8.00 ± 0.05bcA |
| | 25%IWG | | | | 2.98 ± 0.18aA | 7.93 ± 0.02cA |
| | 0%IWG | | | | 2.64 ± 0.09aB | 8.21 ± 0.01aA |
| N1 | 100%IWG | 49.2 ± 1.66A | 49.2 ± 1.66aA | | 4.34 ± 0.57aAB | 7.88 ± 0.04bB |
| | 75%IWG | 43.5 ± 2.15B | 42.0 ± 1.83aB | 1.50 ± 0.33b | 3.58 ± 0.03aA | 7.86 ± 0.03bBC |
| | 50%IWG | 47.0 ± 2.39A | 43.7 ± 2.85aA | 3.34 ± 0.51b | 3.28 ± 0.44aB | 7.95 ± 0.01bA |
| | 25%IWG | 52.4 ± 5.20A | 43.8 ± 6.47aA | 8.66 ± 2.12b | 3.19 ± 0.20aA | 7.96 ± 0.07bA |
| | 0%IWG | 58.4 ± 2.73A | | 58.4 ± 2.73a | 3.51 ± 0.42aAB | 8.39 ± 0.03aA |
| N2 | 100%IWG | 56.6 ± 0.27A | 56.6 ± 0.27aA | | 5.64 ± 0.87aA | 7.78 ± 0.02aB |
| | 75%IWG | 59.2 ± 1.90A | 57.4 ± 2.35aA | 1.82 ± 0.51d | 4.01 ± 0.43aA | 7.73 ± 0.05aC |
| | 50%IWG | 51.3 ± 2.70A | 46.2 ± 2.34bA | 5.08 ± 0.46c | 4.24 ± 0.28aAB | 7.78 ± 0.04aA |
| | 25%IWG | 60.0 ± 1.08A | 51.2 ± 1.68abA | 8.85 ± 0.90b | 4.63 ± 0.40aA | 7.93 ± 0.16aA |
| | 0%IWG | 56.4 ± 0.71A | | 56.4 ± 0.71a | 3.84 ± 0.22aAB | 7.94 ± 0.23aA |
| N3 | 100%IWG | 55.4 ± 2.52A | 55.4 ± 2.52aA | | 5.64 ± 0.14aA | 7.92 ± 0.08aB |
| | 75%IWG | 51.5 ± 2.63AB | 49.1 ± 3.10abAB | 2.36 ± 0.50c | 4.76 ± 0.40aA | 7.92 ± 0.03aAB |
| | 50%IWG | 56.5 ± 2.25A | 51.0 ± 3.06abA | 5.43 ± 0.81c | 5.16 ± 0.27aA | 7.91 ± 0.10aA |
| | 25%IWG | 52.0 ± 1.49A | 41.5 ± 2.09bA | 10.6 ± 0.71b | 4.75 ± 0.77aA | 8.17 ± 0.02aA |
| | 0%IWG | 54.1 ± 1.86A | | 54.1 ± 1.86a | 4.21 ± 0.29aA | 8.12 ± 0.17aA |
| **F-statistic** | | | | | | |
| **Source of variation** | | | | | | |
| N | | 9.75 ** | 8.10 ** | 0.01 | 23.3 *** | 7.61 *** |
| IRF | | 2.34 | 4.73 * | 1252 *** | 4.47 ** | 6.57 *** |
| N*IRF | | 3.10 * | 1.90 | 1.50 | 0.82 | 1.80 |

Notes: Different lower-case letters indicate significant differences at $p < 0.05$ among IRF under the same N level, and different upper-case letters indicate significant differences at $p < 0.05$ among N levels under the same IRF (Tukey's post hoc test). IRF means the species-relative frequency of IWG. Asterisks indicate significant differences, where * indicate $p < 0.05$, ** $p < 0.01$, and *** $p < 0.001$.

## 4. Discussion

### 4.1. Dry Matter Production, Complementary Interactions, and RYT

All values of RYT were larger than one in our study indicated that intercropping of IWG and WC has yield advantages under all species-relative frequencies. For most IWG intercrops, the shoot and root dry matter were similar to that of IWG sole crops, although the relative frequencies of IWG in intercropping were lower than in sole cropping. The 25% IWG intercrops produced the same yields as 100% IWG at N0 and N1 indicating that IWG has a high relative fitness and maintained a high total dry matter production even at low relative frequencies. This result supports the findings reported by Hunter et al. [42] that lower planting density in terms of winder row spacing tended to increase the mean grain yield of IWG. Although in a mixed intercropping system, the dry matter of IWG was not negatively affected by the interspecific competition from WC intercrops. As the result of the competitive ratio showed ($CR_{IWG} > 1$, $CR_{WC} < 1$), the competitive ability of WC was always much lower than that of IWG, and it has not been affected by species-relative frequency or N fertilizer rates. The results of N accumulation and fertilizer N recovery of IWG intercrops also showed that a comparable amount of N with that in IWG sole crop

was accumulated in IWG intercrops despite low IWG-relative frequency. Our results suggest the improvement of dry matter yield and N content of IWG should not rely on overcrowding in sole cropping but the exploitation of complementarity and beneficial interactions between IWG and WC intercrops.

The shoot and root dry matters of WC intercrops were lower than WC sole crops but increased with the decrease in IWG-relative frequency within a specific N fertilizer rate, resulting in an upward tendency of system total dry matter of IWG and WC intercrops. The intercropping advantages of IWG and WC (RYT > 1) in this study were credited to the complementary use of N sources and N transfer from WC to IWG. Under N1 fertilizer condition, WC intercrops fulfilled their N requirement (%NA > 80%) by symbiotic $N_2$ fixation and saved the soil N for IWG intercrops (%NS > 50%), and an average of 12.3% N in IWG intercrops was transferred from WC intercrops. Moreover, the %NA of WC increased from 52.4% in sole crop to an average of 84.0% when intercropped with IWG at N1 due to the high competition of IWG for soil mineral N. These results once again confirmed the widespread theories about the mechanism of intercropping advantages: the complementary use of different N sources by cereal and legume intercrops in low input cropping systems [43], legumes facilitate the growth of associated cereals by transferring N [44], and cereals stimulate $N_2$ fixation of legumes through competition for mineral N in the rhizosphere [27].

### 4.2. Use of Different Nitrogen Sources

Nitrogen accumulations followed the pattern of dry matter yields. The N accumulation of IWG increased with the increase in N fertilizer rates, N accumulation of WC increased with the decrease in IWG-relative frequency, and total N accumulation was affected by the positive interaction of N fertilizer and species-relative frequency. The highest total N accumulation of IWG and WC intercrops existed in 25% IWG with the N3 fertilizer level. In IWG and WC intercropping, IWG intercrops recovered a more than proportional share of fertilizer N sources (more than 40%) in intercropping due to the highly competitive ability, while WC recovered less than 11% of the fertilizer recovery. A similar result reported by Jensen [27] in barley and pea intercrops that the higher competitive ability of barley resulted in the recovery of fertilizer N in the pea to be less than 10% of the total fertilizer N recovery. The highly competitive ability of IWG for fertilizer N forced WC intercrops more relying on the N derived from the atmosphere. We detected that an average of 70.9% of N in all WC intercrops derived from air, only an average of 14.7% derived from fertilizer, and an average of 14.4% from soil under three N fertilizer levels. However, for WC sole crops the proportion of N derived from the air was only an average of 39.7% under three N fertilizer levels, indicating that intercropping with IWG enhanced the proportion of N derived from the atmosphere in WC intercrops, correspondingly reduced the N derived from soil and fertilizer. Different species-relative frequency did not affect the proportion of N derived from air, but N fertilizer application inhibited symbiotic $N_2$ fixation of WC, meanwhile increasing the proportion of N absorbed from fertilizer. The proportion of N derived from the atmosphere in WC intercrop decreased from an average of 84.0 to 57.9% with the increase in N fertilizer level from N1 to N3, and the proportion of N derived from the fertilizer increased from 5.45 to 23.9%. This was consistent with results from Ledgard and Steele [45] who reported that if soil inorganic N was abundant, clover took up relatively more soil N and the proportion of N derived from the atmosphere decreased.

For all IWG intercrops, N came mainly from fertilizer (an average of 42.5%) and soil (an average of 47.6%), only a small proportion, about 10.1% on average, came from apparent N transfer from WC under three N fertilizer levels. The result of measured apparent transferred N varies between different crop stages, measurements, and environmental conditions. Høgh-Jensen and Schjoerring [46] found that the average amount of N transferred from clover to ryegrass was equivalent to 3, 16, and 31% of the N accumulated in ryegrass in the first, second, and third production year. In a split root experiment, Jensen

[27] found that barley obtained up to 19% of its N from intercropped pea when grown in association for 70 days in a soil with a low inorganic N content. Values ranging from 6 to 80% of total N in the grass have been published for N transfer from the legume to the associated grass [47]. Transfer of N from WC to IWG can occur via decomposition of legume root tissues and uptake of the released N by cereal, exudation of soluble N compounds by legumes and uptake by cereal, and transfer of N mediated by plant-associated mycorrhizae [37,48,49]. The transfer of N is mostly long term, as suggested by Jørgensen et al. [36]. In this study, 10.1% N of IWG transferred from WC after 136 days of growth, the potential for transfer is expected to be much higher on a longer time scale.

*4.3. The Role of a White Clover Service Crop on Future N Supply*

Intercropping of cereals and legume service crops is a good strategy to improve N supply and reduce the input of new N fertilizer, with the benefit of $N_2$ fixation and potential transfer of N. In our study, the white clover provided sufficient N to the IWG intercrop to achieve an average of 33.9 g pot$^{-1}$ shoot and root total dry matter without N fertilization. White clover as a service crop can be a relevant contributor to IWG N nutrition and better growth. The amount of $N_2$ fixed was high due to the high %NA, even though the dry matter yields of WC were relatively low. The amount of $N_2$ fixed was correlated with the dry matter yield of WC, as observed in other investigations in clover and ryegrass mixtures [50], indicating that optimum growth conditions could contribute to high dry matter production of WC and further enhance the amount of N fixed. White clover was a weak competitor for inorganic N in intercropping of IWG and WC due to the ability of symbiotic $N_2$ fixation. White clover intercrop was also a weak competitor for light due to short height and shading from IWG at all relative frequency, as Kendall and Stringer [51] reported that the relative growth rates of clover plants decreased rapidly in response to shading. Our previous study [14] showed that alfalfa was very aggressive when intercropped with IWG. We suggest that white clover is a more suitable companion leguminous intercrop for IWG as compared to alfalfa. The intercrops of 75% WC with 25% IWG (3:1) is an optimum combination with relatively low interspecific competition, high amount N fixed, and high RYT in this study.

Furthermore, when we calculated how much N could be fixed per g of WC, we found that 1 g WC dry matter contributed an average of $11.8 \times 10^{-3}$ g fixed N to the intercropping systems, which was equal to 11.8 kg N t$^{-1}$ WC dry matter. This value was slightly higher than the reports from Hayes et al. [11] of subterranean clover (<10 kg N t$^{-1}$ dry matter), probably because N fixed by WC roots was also included in the $N_2$ fixation in our study. When we compared the N accumulation of IWG intercrops at N0 with N accumulation of IWG sole crops at N1, we found that despite the low species-relative frequency, the IWG intercrops achieved comparable N accumulation (from 0.36 to 0.46 g pot$^{-1}$) at N0 to the IWG sole crop (0.70 g pot$^{-1}$) fertilized with 75 kg N ha$^{-1}$, suggesting that the WC service crop can supply enough N for IWG under appropriate soil N fertility and species-relative frequencies.

**5. Conclusions**

This study showed that IWG and WC intercrops have the potential to improve the use efficiency of N source and land productivity due to competitive, facilitative interactions, complementary use of soil mineral N and atmospheric $N_2$, and N transfer from WC to IWG. The intercrops of IWG, which have a highly competitive ability for N, acquired a much larger proportion of soil and fertilizer N, consequently forcing WC intercrops more relying on the N derived from the atmosphere. Decreasing IWG-relative frequency from 75 to 25% did not affect the %NA, %NT, %NF, %NS, RYT, dry matter, and N accumulation of IWG, while increased dry matter and N accumulation of WC, resulting in the increases in amounts of $N_2$ fixed, total dry matter and N accumulation in IWG and WC mixed intercropping. The incremental levels of N fertilizers increased %NF of both WC and IWG, resulting in decreased %NA of WC and decreased %NS of IWG, indicating that white

clover would rely more on N in fertilizers than on symbiotic N₂ fixation if an excessive amount of N fertilizer was applied, which could impair the complementary effect in IWG and WC intercrops, resulting in inefficient utilization of N resources. White clover as a service crop could supply sufficient N for IWG intercrops under appropriate soil N fertility and species-relative frequencies.

**Author Contributions:** Conceptualization, S.L., E.S.J., L.-M.D.M., and Y.Z.; methodology, S.L. and E.S.J.; software, S.L.; formal analysis, S.L.; investigation, S.L.; resources, L.-M.D.M. and N.L.; data curation, S.L.; writing—original draft preparation, S.L.; writing—review and editing, E.S.J., L.-M.D.M., Y.Z., and N.L.; supervision, Y.Z. and L.-M.D.M.; project administration, L.-M.D.M.; funding acquisition, Y.Z. All authors have read and agreed to the published version of the manuscript.

**Funding:** This research was funded by the China Forage and Grass Research System (CARS-34).

**Institutional Review Board Statement:** Not applicable

**Informed Consent Statement:** Not applicable

**Data Availability Statement:** Not applicable

**Acknowledgments:** The authors gratefully acknowledge the Land Institute for supplying the seeds of intermediate wheatgrass used for experiments. We would like to thank Jan-Eric Englund for help with data analysis. We are grateful to Yuqi Wei, Tao Li, and Ye Wu (China Agricultural University) for helping us collect plant and soil samples. We also thank the anonymous reviewers for their careful reading of our manuscript and their many insightful comments and suggestions.

**Conflicts of Interest:** The authors declare no conflict of interest.

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
