# Peer review of "Species Interactions and Nitrogen Use during Early Intercropping of Intermediate Wheatgrass with a White Clover Service Crop"

_agronomy, doi:10.3390/agronomy11020388_

Round 1

Reviewer 1 Report

No further suggestions

Author Response

Thank you for reviewing our manuscript and thank you for your suggestions and comments! 

Reviewer 2 Report

Dear authors

This paper is definitely in an area of interest and need.

The authors need to acknowledge the problems behind using a non-fixing reference crop to measure BNF, as well as how tricky is to use isotope dilution method. Particularly acknowkledging that 15N-enrichment will be diluted as soon as it is added to the system, thus, its presence in the system differs completely from a regular N fertilizer 14N.

The main highlighst from this review is that methods need to be reorganized and summarize certain parts, reinforce the strength and weakness of isotope dilution. In the case of their field or greenhouse activities, they can create a table, in that way they could save lots of space in the materials and methods section. Regarding calculations, there is a need to state that all equations will be used for each treatment (N level x relative frequency).

Regarding the statistical analysis, I think that the authors made a big mistake in using all 5 replicates of their experiment and treat them as similar when doing statistical analysis. The authors should have kept separately replicates that were 15N-enriched from the others none-enriched, even though authors aim to add similar N amount. In the statistical analysis section the authors mentioned that (Line 288-289) that they have justified this strategy through the equivalence test. I did not find it this ‘equivalence test’ in the manuscript, nor a robust explanation why they lumped all replicates from different treatments as if they were one.  I will be careful in doing this lumping, because you are assuming that 15N fertilizer behaves similarly as 14N, and that, it is not true, for that reason is called isotope dilution. The 15N gets super quickly diluted with the 14N of the environment, in this case, soil organic matter.

 I believe that this paper needs a little bit of more polishing and reinforcement in the highlighted parts. Below you will find some corrections for each section.

Abstract: about using acronyms for both treatments IWG and WC, be consistent throughout the manuscript.

Line 62-63: unclear sentence, please reworded.

Line 71: week competition?

Question, expand more about What is the frequency? And why is important.

Cover the previous literature related to the topic. Suggestion expands line 99-103.

Highlight why this project is important and which specific research gap is trying to cover.   

Line 150: please explain why did you split the N fertilizer application into 3 dates?

Line 153: why did you decide to add KCL to your N fertilizer additions?

Line 153: When you are working with isotope dilution

Line 114: I will recommend including a table with all the activities, for instance, installation of greenhouse experiment Feb 15, Application 1, App 2, Appl 3, Sampling, Harvest July 2. In this way, you can skip placing this in the methods.

Line 168: How did you know which roots belong to IWG or WC? How did you make sure that you recovered almost all root biomass from each crop?

Line 177: which methodology did you use to determine inorganic N? please cite your method.

Line 205: Why do you use italics in “That is”?

Line 203: What do you refer as “sole”?

Line 185-232: I suggest reorganizing all the calculation process because is somewhat disorganize and hard to follow. It is hard to follow which treatments were used on these calculations. I suggest expanding our literature review, perhaps checking out Unkovich et al (2008). Question: Where did you use your non-labeled fertilized replicated pots? Did you correct your actual 15N enrichments in your labelled samples by subtracting the average value from your control pots that were not labelled?

Line 219-225: Would you mind citing the source of these calculations of the transport of N from IWG mix to WC? I am afraid that there is no transfer while both plants are alive. I think that the 15N labelled fertilizer will be uptake by the two plants at different proportions due to different root architecture and root N uptake rate. Thus, I will call this %Nuptake split within the intercropping treatment.

Line 224: which N level 15N enrichment did use for your calculations of NTshoot?

Line 263: I suggest you change the word “sole” for “control”, so you make it clear that you are referring to the treatment of the 0N from each plant.

Author Response

This manuscript is a resubmission of an earlier submission. The following is a list of the peer review reports and author responses from that submission.

Round 1

Reviewer 1 Report

Dear authors,

the experiment is nicely described. Please spend the same effort in describing your statistical methods as in describng the experiment. I gave several suggestions to enhence the quality of the manuscript. Please take your time to account for these.

Author Response

Response to Reviewer 1 Comments

Point 1: The paper “Species interactions and nitrogen use during early intercropping of intermediate wheatgrass with a white clover service crop” presents result from a greenhouse experiment. A CRD was used to test the influence of the amount of fertilizer on five different intercropping and sole cropping treatments. The description of the experiment performed is fine and understandable, thank you for this. The paper is well written even if there are several mistakes inside. Unfortunately, the statistical analysis and the calculations done with the measurements remains unclear. The authors seem to divide their five replicates in two subgroups and varied the fertilizer between groups by adding N15 or not.  This has the advantage that the authors can test for identity of both fertilizer versions, but this test is not done. Furthermore, all equations lacks a proper use of indices so that it is unclear, which observations (or means?) were used, and how this relates to the ANOVA or generalized linear model.

Response 1:

Thank you for your suggestions and comments! A detailed description of statistical analysis has been added, see L266-277. The two types of fertilizers used in this study are similar in all other ways except for the 15N enrichment. It was reported that the development, productivity, and nitrogen use of wheat plants are unaffected by the stable isotope form of the nitrogen source. Subedi and Ma (2010) found that give similar concentrations of the two N treatments that differed markedly in their 15N enrichment, there was no differentiation in N uptake and partitioning resulting from isotopic enrichment of the two N sources. Based on this, we assumed that the difference of 15N enrichment does not bring any differences in the growth and dry matter yields of plants in this study. Therefore, five replicates were used in the analysis of variance for dry matter yield, root/shoot ratio, N accumulation, relative yield total (RYT), competitive ratio (CR), soil mineral N, and PH, since five replicates gave better robustness than three replicates. Three replicates were used in the analysis of variance for the proportion of N derived from atmosphere (%Na), fertilizer (%Nf), soil (%Ns), transfer (%Nt), the amount of N2 fixed, N transferred, and the fertilizer N recovery, because only three of five replicates were fertilized with 15N fertilizer, and the rest two replicates were fertilized with unlabeled fertilizer.

Major comments:

Point 2: 1. Clarify which observations were used in which equation. Please add indices for both factors and the replicate. Further note that a standardization with a mean estimates from two pots causes a correlation of the data the authors had not accounted yet. The same holds for the RYI analysis (which is not correct at the moment!). In this case, the authors can do the analysis with the raw values and calculate the ratio/difference later on. Note that for log-normal distributed data the difference on the log-scale corresponds to the ratio on the original scale.

Response 2: A detailed description of each equation has been added, see L174-263.

What do you mean “a mean estimates from two pots causes a correlation of the data”? The mean value is the average of five or three replicates. Do you mean the dry matter yield of IWG and WC has a correlation because they came from the same pot?

The calculation of RYT is right because it is actually not a percentage, but a well-known expression used in the area of intercropping, and ANOVA has been used to analyze RYT in numerous published papers.

Point 3: 1. The model for analysis is unclear. If the authors used a generalized linear model, they have to specify the link function. In this case please present the linear predictor as equation and whether the authors account for potential over-dispersion. If the authors used a two-way ANOVA, they should specify the model using a model equation adding their assumption of variance. Note that the variances are likely to be heterogeneous as the number of plants per species vary from 4 to 16. Justify your model choice by reporting how you checked model requirements.

Response 3: Two-way ANOVA was performed by a general linear model, not a generalized linear model. What do you mean the variances are likely to be heterogeneous as the number of plants per species varies from 4 to 16? Do you mean the variances of data from 4 plants are larger than 16 plants?

Point 4: The authors used two versions of fertilizer. They should check that they are identical (test of equivalence).

Response 4: It was reported that there was no differentiation in the performances of plants (including in growth, development, and dry matter production) grown with N sources that were either predominantly 14N or 15N enriched (Subedi and Ma, 2010). We don’t think it is necessary to test the equivalence of two versions of fertilizer. The 15N-labeled treatments have three replicates, and unlabeled controls only have two replicates. If two variables have a different number of replicates, can equivalence testing be conducted?

Point 5: The authors reported yield or other traits from both species. Values from the same pot are correlated and the authors can account for the correlation to borrow strength across traits. A bivariate analysis can be used or at least discussed.

Response 5: If we understand correctly, the purpose of using bivariate analysis is to test if one species is low probably the other one is large. This kind of interaction between two species, e.g. the impact of 4 IWG to 12 WC, was calculated by the competitive ratio (CR).

It is difficult to do a bivariate model, maybe the correlation could be calculated within the pot to see if two species are correlated. But we do not think the correlation between 4 IWG and 12 WC, 8 IWG and 8 WC, or 12 IWG and 4 WC is relevant to our research questions.

Point 6: The authors used fertilizer as factor, but it is numeric. The authors can and should check for linear or non-linear relationships e.g. by using some kind of polynomial regression till the quadratic term.

Response 6: Thank you for your suggestions!  Regression can be used to estimate the relationship between N fertilizer level and yield and predict the agronomical optimum nitrogen rates. But this study mainly focused on the effect of N fertilizer, species relative frequency, and their interactions on N2 fixation, N transfer, and N proportion derived from different sources. ANOVA is the most suitable method to analyze these effects, and the best combinations of N fertilizer level and species relative frequency can be selected by ANOVA.

If we look at the figures, it is not obvious that a polynomial regression is well fitted to this data. The number of levels for N is also not enough to find a trustworthy model. 

Besides, a second degree equation has three parameters, considering the numbers of factor, levels, and variables in this experiment, it is impractical to do polynomial regression.

Point 7: The authors reported means with mean-specific SE. I guess (not sure about that) that they calculate both independent from the fitted models, and thus implicitly assumed another model here. Please select an appropriate model and calculate (adjusted) means from this model. If you think variances are not equal, you can account for this within the model.

Response 7: Could you please elaborate a little bit? What do you mean by “thus implicitly assumed another model here”?

Point 8: The variable names are terrible. Furthermore, not all are explained. In most cases one can guess what the authors meant, but it helps the reader to have a clear structure how variables are denoted.

Response 8: The meaning of each variable has been explained in detail, see L175-263. The %Ndfa was replaced by %Na, %Ndff was replaced by %Nf, %Ndfs was replaced by %Ns, %Ndft was replaced by %Nt.

Point 9: From the results it seems to be that 4 IWG plants are producing the same yield as 16. Please justify that the gain from intercropping based on intercropping and not from intra-specific completion. The study is only valid, if a seed density of 16 plants per pot is the optimal plant density for both species.

Response 9: The reasoning related to intraspecific competition has been deleted, and discussion based on my own data has been supplemented, see 407-432. We agree that the results of this study only valid under the condition of 16 plants per pot. The boundary of the research results has been stated, see L417-8.

Point 10: The authors tested the proportion of N. Note that there is a sum to one restriction in the data and thus tests are not independent. Not sure what the authors did, but the proposed normal distribution is properly violated as data are expected to be binomial distributed.

Response 10: The proportion of N derived from the soil in WC was calculated by the equation %Nf + %Ns + %Na = 100%, but the data of %Ns in WC fulfilled the normal distribution since the p-value of Shapiro-Wilk test was larger than 0.05. The %Nf and %Na are percentages but they are a continuous part divided by a continuous part, so we cannot use the binomial distribution here.

Minor comments

Point 11: There are several typos, e.g. 10[space]%, which is fine per definition, but is used without space in English. Don´t use line breaks between -1.

Response 11: Thank you for your suggestions! The space between numbers and % have been deleted. (Although the Swedish standard is to have space between number and %.)

You also mentioned not using line breaks between -1. There is no line break between numbers and %. Which line break do you mean?

Point 12: Tukey test is done under alpha=0.05, and not p.

Response 12: Thank you for correcting this mistake! Changes have been made, see L271.

Point 13: Please correct the description of letter display, as two means with ab and bc letters are not significantly different from each other (in contrast to your description)

Response 13: The description of results in L281-2, L298, and L351-3 was revised.

Point 14: Please report p-values. Note that there is a linear relationship between both.

Response 14: p-values have been reported in the results section, see L280-389. You can also find p-values in tables presented as *,**,***.

Linear relationship between BOTH? Do you mean there is a linear relation between the F-value and the P-value?

References:

Subedi, K. D., Ma, B. L. (2010). Wheat Development, Productivity, and Nitrogen Use Are Unaffected by Stable Isotope Form of the Nitrogen Source. Crop science, 50(4), 1490-1495, doi: 10.2135/cropsci2009.09.0494

Reviewer 2 Report

Manuscript number: agronomy-1013795

Title: Species interactions and nitrogen use during early intercropping of intermediate wheatgrass with a white clover service crop

This paper reports the results of a single experiment mixing in pots intermediate wheatgrass (IWG) and white clover (WC). Mixed crops with WC was supposed providing N to increase the yield of IWG while sparing fertilizer. Various levels of nitrogen fertilizer and IWG relative frequency (IRF) were thus compared for sample harvest at early flowering. Using 15N labeling of fertilizer, the N sources were assessed in both plants, ie labeled N from fertilizer, and unlabeled N from soil and air. Assumptions were however needed to separate N fluxes from soil and from air in WC, and to assess N transfer from WC to IWG. I'm not completely convinced by these assumptions, as described below. Data are properly presented, but the paper was too rapidly written, leading to weaknesses in both form and substance. Literature is not properly combined with data, and hypotheses are sometimes discussed regarding literature instead of data. Finally, errors in calculations, as described below, preclude estimating this paper. I recommend authors more deeply regard their data before consideration.

More detailed review below.

English is sometime confusing (eg. In L185, 356, 371, 423), but fortunately not at key points.

  • In L393-4 and L403 please indicate under which conditions (N-IRF) refer the reported data.
  • According to L399-400, in WC, 'the proportion of N derived from the atmosphere is likely to be substituted by the proportion of N derived from fertilizer, if excessive N fertilizer is supplied'. Yet N content in IWG continuously increased with fertilizer, whereas RYT decreased; therefore excessive in which meaning?
  • According to L420, 'the white clover provided sufficient N to the IWG intercrop to achieve average 33.9 gDM pot-1 without N fertilization'. However, according to tab.1, yield was 32.5 gDM pot-1 in pure IWG under N0. To what refers 33.9 g pot-1?
  • Authors however used too complex acronyms and subscripts. For instance, N content originating from air (equation 1) is termed %Ndfa whereas %Na would be enough. In L222, is that any difference between Ziwg and IRF? If not, one could give up Ziwg. Similarly, Zwc should be 1-Ziwg; therefore, you could omit Zwc. Lastly, according to their definitions (equations 12 &13), CRi should be 1/CRw. However, data in table 1 aligned instead to CRi #1.1 CRw. Please check your calculations, something sounds wrong.

Miscellaneous:

  • According to L116, available potassium in soil amounted 120 kg/ha. Moreover, K was always associated to N in fertilizer: up to 750 kg/ha. Therefore, I disagree with statement in L152 that 'No phosphates or potash fertilizers were applied in this experiment'.
  • According to L118-119 the deepness of pots was 8 cm only, which is wise for a labelling experiment, but very few to characterize IWG as a deep-rooted crop (L360). In L376-380, Authors recognize the competition IWG vs WC was thus biased. I feel the reasoning in L366-368 is therefore invalid. Yet the section could be saved if built on authors own data instead of literature compliance.
  • In L349, authors wrote: ‘IWG and WC mixed intercrops have yield advantages (RYT>1)’. Yes, but tab1 reports RYT at 1.01±02. Yield advantage is demonstrated if RYT is significantly >1, which needs to be checked.
  • L369 claims that 'the relative growth rate of individual IWG increased at a low relative frequency'. Authors do not have any data about that, they just make a hypothesis.
  • According to L386, 'the highest total N accumulation existed in 25 % I with the N3 fertilizer level'. Wrong, the highest total N accumulation occurred in WC pure strand. Unlike previously stated by authors, N did not exactly follow DM, as intercropping decreased N while increasing DM.
  • L424 states that 'the amount of N2 fixed was correlated with the total dry matter yield, as observed in other investigations in clover and ryegrass mixtures'. I don't know for literature but regarding the present experiment, the amount of fixed N is reported in Tab.2, and the DM yield in Tab.1: no correlation at all between these data.
  • In L439, authors wrote 'the IWG intercrops achieved comparable N accumulation (average 0.42 g pot‑1) to the IWG sole crop (0.66 g pot-1) fertilized with 75 kg N ha-1'. Well, Tab.2 doesn’t report any IWG N in intercrop at 0.42 g pot‑1, and I'm unable to follow author's reasoning. Instead, as the best intercrop was [N1; IRF 25%] according to authors (L342), it should be compared to IWG pure strand [N1; IRF 100%] yielding 0.66 gN pot-1. The [N1; IRF 25%] intercrop yielded 0.74 gN pot-1 in IWG according to tab.2. According to authors claim in introduction, they should respond to the following questions: Is this difference significant? Is it worth?
  • According to their accuracy, too much digit burden some numbers reported in tables. For instance please write in table 1: total yield =33±1 and not 32.5±43; CRi = 11±5 and not 10.7±4.99.

Main problems

  • I was puzzled with some assumption for N fluxes assessment. For instance Equation (1) assess N fluxes from air to WC using IWG as a reference despite IWG response to fertilizer clearly differs from that of WC, as paper shows it. Otherwise, I do not understand how equation (4) takes into account the fact that N transferred from WC to IWG is a mix of labelled and unlabeled N, which could bias the results according to fertilizer level. In any case, the burden of measurements error is fully reported on transfer fluxes, which could explain authors failed to detect any consistent response of transfer fluxes to experimental conditions.
  • According to Tab.3 and to data elsewhere, the recovery of N fertilizer was very low: about one half only. However, mineral N almost vanished in pots, suggesting a huge level of organic N was built in only 136 days, whereas mineral N was extracted! Moreover, the N contents in plant were not explained by the low fertilizer contribution. According to fig.2 in IWG pure strand fertilized at N3 (225 kg ha-1) N from fertilizer was only one-half of N in plant which is quite incredible. Fig.2 therefore supposed the remaining N in plants originate from soil N. However, according to L115, the initial soil N was 0.54 g pot-1, i.e. less than the calculated flux to plants! In other words, soil is supposed providing more that its initial content of unlabeled N while accumulating labeled N at the same level. Moreover N providing by soil needs a complete mineralization of its initial N, while synchronously N accumulates in organic form only… Clearly, something is wrong. You should revise your calculations before consideration.

Author Response

Response to reviewer 2 comments

Point 1: This paper reports the results of a single experiment mixing in pots intermediate wheatgrass (IWG) and white clover (WC). Mixed crops with WC was supposed providing N to increase the yield of IWG while sparing fertilizer. Various levels of nitrogen fertilizer and IWG relative frequency (IRF) were thus compared for sample harvest at early flowering. Using 15N labeling of fertilizer, the N sources were assessed in both plants, ie labeled N from fertilizer, and unlabeled N from soil and air. Assumptions were however needed to separate N fluxes from soil and from air in WC, and to assess N transfer from WC to IWG. a. 1. I'm not completely convinced by these assumptions, as described below.  Data are properly presented, but the paper was too rapidly written, leading to weaknesses in both form and substance. 2. Literature is not properly combined with data, and hypotheses are sometimes discussed regarding literature instead of data. 3. Finally, errors in calculations, as described below, preclude estimating this paper. I recommend authors more deeply regard their data before consideration.

Response 1:

Thank you for your comments!

  1. A detailed description of the principles of the 15N isotopic method has been added to the manuscript, see L175-180. You can also find the description in response 15 since these two are the same question.
  2. Discussion on shallow- and deep-rooted plants, intraspecific competition, and relative growth rate verified your opinion that hypotheses are sometimes discussed regarding literature instead of data. These discussions have been deleted (L434-459). Reasoning based on data was added, see L409-418, L423-432.
  3. According to the below comments (Points 6, 13, 16), the reasons why you think errors existed in calculations were: CRI×CRw≠1, some data used in the discussion cannot be found in tables, and soil provided more than its initial content of N to plant.

CRI×CRw≠1 is because the data presented in table 1 are the average values of five replicates. The product of CRI and CRw for every individual repetition exactly equals 1. (Response 6).

Some data like 0.42 g pot-1 cannot be found in Table 2, because it is the average value of N accumulation of three IWG intercrops (75% I, 50% I, and 25% I) at N0. (Response 13)

The initial soil N was 0.54 g kg-1, not 0.54 g pot-1! Every pot has 10 kg of soil, so the amount of initial soil N was 5.4 g pot-1. The N provided by the soil to the plants did not exceed the initial N in the soil. (Response 16)

More detailed review below.

Point 2: English is sometime confusing (eg. In L185, 356, 371, 423), but fortunately not at key points.

Response 2: L185 has been revised, see L193-199. L356 has been deleted. L371 has been rewritten, see L419-421. L423 has been revised, see L504-5.

Point 3: In L393-4 and L403 please indicate under which conditions (N-IRF) refer the reported data.

Response 3: Numbers in L393-4 are the average value of all WC intercrops under all N fertilizer levels. The conditions are specified. See L471-4 and L485-7.

Point 4: According to L399-400, in WC, 'the proportion of N derived from the atmosphere is likely to be substituted by the proportion of N derived from fertilizer, if excessive N fertilizer is supplied'. Yet N content in IWG continuously increased with fertilizer, whereas RYT decreased; therefore excessive in which meaning?

Response 4: What I was trying to say here is that WC may completely dependent on N from fertilizer and stop symbiotic N2 fixation if excessive N fertilizer is supplied. This is a speculation, and excessive N fertilizer is also a vague concept, so the original sentence has been deleted, and the sentence in L478-481 has been added for supporting the argument in the previous sentence.

Point 5: According to L420, 'the white clover provided sufficient N to the IWG intercrop to achieve average 33.9 gDM pot-1 without N fertilization'. However, according to tab.1, yield was 32.5 gDM pot-1 in pure IWG under N0. To what refers 33.9 g pot-1?

Response 5: 33.9 g DM pot-1 is the average yield of all IWG intercrops under N0. White clover intercrops provided sufficient N to the IWG intercrops to achieve an average 33.9 g DM pot-1 without N fertilization. 33.9 g DM pot-1 can only be calculated by the data from Figure 1. The yield 32.5 g DM pot-1 you mentioned is the total yield of IWG sole crop under N0. White clover cannot provide N for IWG sole crop since they were not planted together. So L420 has not changed, now it is L501-3.

Point 6: Authors however used too complex acronyms and subscripts. 1. For instance, N content originating from air (equation 1) is termed %Ndfa whereas %Na would be enough. 2. In L222, is that any difference between Ziwg and IRF? If not, one could give up Ziwg. Similarly, Zwc should be 1-Ziwg; therefore, you could omit Zwc. 3. Lastly, according to their definitions (equations 12 &13), CRi should be 1/CRw. However, data in table 1 aligned instead to CRi #1.1 CRw. Please check your calculations, something sounds wrong.

Response 6: Thank you for your suggestions! 1. The acronyms and subscripts are terrible in this article. The %Ndfa was replaced by %Na, %Ndff was replaced by %Nf, %Ndfs was replaced by %Ns, %Ndft was replaced by %Nt according to your suggestions.

  1. There is no difference between ZIWG and IRF, so the ZIWG was replaced by IRF, and ZWC was changed to 1- ZIWG. See line 259.
  2. According to equations 12 and 13, CRI should be 1/CRW and it is for each individual repetition. In each repetition, the product of the CRs of IWG and WC is one. But the CRI or CRW under specific N-IRF presented in table 1 is the mean value of five replicates. The product of mean values is not exactly equal to one.

Miscellaneous:

Point 7: According to L116, available potassium in soil amounted 120 kg/ha. Moreover, K was always associated to N in fertilizer: up to 750 kg/ha. Therefore, I disagree with statement in L152 that 'No phosphates or potash fertilizers were applied in this experiment'.

Response 7: Thank you for your comments! It is incorrect to say that no potash fertilizer was applied in this experiment, not only because the available potassium in base soil is high, but also KCl was applied to 15N labeled treatments and KNO3 was applied to unlabeled controls. This sentence 'No phosphates or potash fertilizers were applied in this experiment' has been deleted. (L152)

Point 8: According to L118-119 the deepness of pots was 8 cm only, which is wise for a labelling experiment, but very few to characterize IWG as a deep-rooted crop (L360). In L376-380, Authors recognize the competition IWG vs WC was thus biased. I feel the reasoning in L366-368 is therefore invalid. Yet the section could be saved if built on authors own data instead of literature compliance.

Response 8: According to L118-119, the deepness of pots was 26.5 cm, not 8 cm. Even in a pot, IWG produced a considerable amount of roots and WC produced fewer roots than IWG. My argument in L376-380 is that due to the limit of pot volume, soil depths did not lead to strong root differentiation in mixed intercrops, resulting in less complementarity in soil resource utilization compare to intercrops in the field. But as you said this reasoning built on literature compliance instead of my own data, so L434-459 has been deleted, and new discussion has been supplemented, see L409-432.

Point 9: In L349, authors wrote: ‘IWG and WC mixed intercrops have yield advantages (RYT>1)’. Yes, but tab1 reports RYT at 1.01±02. Yield advantage is demonstrated if RYT is significantly >1, which needs to be checked.

Response 9: What is the definition of SIGNIFICANTLY large than one? The original article did not state clearly that yield advantage only exists if RYT significantly large than one (Wit and Bergh, 1965). The lowest RYT in this experiment is 1.01±0.02, indicating that the area planted to sole crops would need to be 1% greater than the area planted to the intercrops for the two to produce the same dry matter yield. The largest RYT 1.39±0.08 indicating that sole crops need 39% greater area than intercrops to produce the same yield. I would say yield advantage exists even if RYT slightly higher than one, and it is not wrong to state ‘IWG and WC mixed intercrops have yield advantages (RYT>1)’.  No changes have been made in L404.

Point 10: L369 claims that 'the relative growth rate of individual IWG increased at a low relative frequency'. Authors do not have any data about that, they just make a hypothesis.

Response 10: The data of dry matter of IWG individual plant (g plant-1) was analyzed but did not present in the manuscript, since it will duplicate with the dry matter yield per pot. The results of IWG yield per plant show that individual IWG plant has high yields at a low relative frequency so that 4 plants can maintain the same total yields with 16 plants. Individual IWG plant has a high yield (g plant-1) at low relative frequency can also be inferred from the constant total yield per pot. However, it is not accurate to state the relative growth rate of individual IWG increased at a low relative frequency since I did not measure the growth rate directly, so this sentence has been deleted.

Point 11: According to L386, 'the highest total N accumulation existed in 25 % I with the N3 fertilizer level'. Wrong, the highest total N accumulation occurred in WC pure strand. Unlike previously stated by authors, N did not exactly follow DM, as intercropping decreased N while increasing DM.

Response 11: Thank you for pointing out this mistake! What I meant to say was that “the highest total N accumulation of IWG and WC intercrops existed in 25 % I with the N3 fertilizer level”. Changes have been made, see L464.

Point 12: L424 states that 'the amount of N2 fixed was correlated with the total dry matter yield, as observed in other investigations in clover and ryegrass mixtures'. I don't know for literature but regarding the present experiment, the amount of fixed N is reported in Tab.2, and the DM yield in Tab.1: no correlation at all between these data.

Response 12: There is a correlation between the amount of fixed N by WC and the dry matter of WC because the amount of fixed N is calculated by the dry matter yield of WC. N fixed = (WC dry matter × %N in WC/100 × %Ndfa/100). The problem here is I did not clarify that it is the dry matter of WC which correlated to N2 fixation, not the total dry matter yield in Table 1. Thank you for pointing out this mistake! Changes have been made, see L507.

Point 13: In L439, authors wrote 'the IWG intercrops achieved comparable N accumulation (average 0.42 g pot‑1) to the IWG sole crop (0.66 g pot-1) fertilized with 75 kg N ha-1'. 1. Well, Tab.2 doesn’t report any IWG N in intercrop at 0.42 g pot‑1, and I'm unable to follow author's reasoning. Instead, as the best intercrop was [N1; IRF 25%] according to authors (L342), it should be compared to IWG pure strand [N1; IRF 100%] yielding 0.66 gN pot-1. 2. The [N1; IRF 25%] intercrop yielded 0.74 gN pot-1 in IWG according to tab.2. According to authors claim in introduction, they should respond to the following questions: Is this difference significant? Is it worth?

Response 13: 1. The IWG intercrops achieved comparable N accumulation (average 0.42 g pot‑1) to the IWG sole crop (0.66 g pot-1) fertilized with 75 kg N ha-1. 0.42 g pot‑1 cannot be found in table 2 because it is the average value of all IWG intercrops at N0. To make it clearer, the sentence has been changed the IWG intercrops achieved comparable N accumulation (from 0.38 to 0.44 g pot-1) at N0 to the IWG sole crop (0.66 g pot-1) fertilized with 75 kg N ha-1. (See L524)

  1. The difference between 0.74 g N pot-1 and 0.66 g N pot-1 is insignificant statistically, and that is exactly the interesting part, because 25% IWG in intercropping achieved a similar amount of N accumulation with 100% IWG in sole cropping under N1.

Point 14: According to their accuracy, too much digit burden some numbers reported in tables. For instance please write in table 1: total yield =33±1 and not 32.5±0.43; CRi = 11±5 and not 10.7±4.99.

Response 14: The data format of total yield and CRI could be modified as your suggestions, but what about the rest of the data in table 1? Should I change all data formats for consistency? If so, all CRW are 0, and all RYT is 1. 

Main problems

Point 15: I was puzzled with some assumption for N fluxes assessment. 1. For instance Equation (1) assess N fluxes from air to WC using IWG as a reference despite IWG response to fertilizer clearly differs from that of WC, as paper shows it. 2. Otherwise, I do not understand how equation (4) takes into account the fact that N transferred from WC to IWG is a mix of labelled and unlabeled N, which could bias the results according to fertilizer level. 3. In any case, the burden of measurements error is fully reported on transfer fluxes, which could explain authors failed to detect any consistent response of transfer fluxes to experimental conditions.

Response 15: 1. Legume assimilates both atmospheric N2 and soil mineral N. Soil is often slightly higher in 15N abundance than is atmospheric N2. This small difference can be utilized to distinguish between legume N originating from the soil and the sir. Artificially enriched in 15N fertilizer can be added to the soil to expand the difference in the 15N compositions of soil N and atmospheric N2. The low 15N abundance of atmospheric N2 “dilutes” the higher 15N concentration of soil-derived N in legume as atmospheric N2 is fixed. The 15N enrichment of non N2-fixing reference plants should accurately reflect the 15N enrichment of soil N taken up by the legume. The amount of N taken up from the soil by the reference plants does not have to be the same as that for the legume. (Unkovich et al., 2008)

  1. Fertilizer levels can influence the amount of N transfer. However, the numerator 15N enrichment of IWG mixed intercrops and the denominator 15N enrichment of IWG mixed intercrops IWG sole crops in equation 4 are under the same N fertilizer level, and the result is the N transfer at a given N level.
  2. The burden of measurement error indicates no consistent N transfer fluxes? The standard errors of N transfer are similar to standard errors of other parameters in table 2. The low values of N transfer make its standard errors looks relatively high.

Point 16: According to Tab.3 and to data elsewhere, 1. the recovery of N fertilizer was very low: about one half only. 2. However, mineral N almost vanished in pots, suggesting a huge level of organic N was built in only 136 days, whereas mineral N was extracted! Moreover, the N contents in plant were not explained by the low fertilizer contribution. According to fig.2 in IWG pure strand fertilized at N3 (225 kg ha-1) N from fertilizer was only one-half of N in plant which is quite incredible. Fig.2 therefore supposed the remaining N in plants originate from soil N. 3. However, according to L115, the initial soil N was 0.54 g pot-1, i.e. less than the calculated flux to plants! In other words, soil is supposed providing more that its initial content of unlabeled N while accumulating labeled N at the same level. Moreover N providing by soil needs a complete mineralization of its initial N, while synchronously N accumulates in organic form only… Clearly, something is wrong. You should revise your calculations before consideration.

Response 16: 1. The total recovery of N fertilizer is from 43.5 % to 60.0 % which is reasonable according to Unkovich et al. (2008), the efficiency of fertilizer N uptake is commonly 30-60% and plants will also use indigenous soil mineral N.

  1. Why vanish of mineral N must suggest a huge level of organic N was built? The available soil N could be absorbed by plants, lost by ammonia volatilization or denitrification.
  2. The initial soil N was 0.54 g kg-1, not 0.54 g pot-1! Every pot has 10 kg of soil, so the amount of initial soil N was 5.4 g pot-1. The highest total N accumulation of two plants (WC and IWG) is 1.86 g pot-1 which is much less than the initial N in the soil. The N provided by the soil to the plants did not exceed the initial N in the soil.

References:

Wit, C., Bergh, J.V. (1965). Competition between herbage plants. The Journal of Agricultural Science, 13, 212-221.

Unkovich, M., Herridge, D., Peoples, M., Cadisch, G., Boddey, B., Giller, K., Alves, B., Chalk, P. (2008). Measuring plant-associated nitrogen fixation in agricultural systems. Australian Centre for International Agricultural Research (ACIAR).

Reviewer 3 Report

A little too much detail.

Author Response

Point: A little too much detail.

Response: Could you please elaborate a little bit? Which section is too detailed? Do you mean the materials and methods section?

Round 2

Reviewer 1 Report

Dear authors,

the revision still performs wrong statistical analysis, and all major comments were not yet considered. Correcting the statistical anaylsis is (in my point of view) a pre-requirement for a publications. Please avoid to submit another revision without accounting for all my suggestions (or have good aguments not to do so). I really wonder, what you expect in the second round of review, if you are not accounting for reviewers recommendations.

Please find the same comments and responses on your questions in the attachment.
